# GQA-$\mu$P: The Maximal Parameterization Update for Grouped Query Attention

## Abstract

Hyperparameter transfer across model architectures dramatically reduces the amount of compute necessary for tuning large language models (LLMs). The maximal update parameterization ($\mu$P) ensures transfer through principled mathematical analysis but can be challenging to derive for new model architectures. Building on the spectral feature-learning view of Yang et al. (2023a), we make two advances. First, we promote spectral norm conditions on the weights from a heuristic to the definition of feature learning, and as a consequence arrive at the Complete-P depth and weight-decay scalings without recourse to lazy-learning. Second, we consider a modified spectral norm that preserves the valid scaling law of network weights when weight matrices are not full rank. This enables (to our knowledge, the first) derivation of $\mu$P scalings for grouped-query attention (GQA). We demonstrate the efficacy of our theoretical derivations by showing learning rate transfer across the GQA repetition hyperparameter as well as experiments regarding transfer over weight decay.

## 1 Introduction

The maximal update parametrization, or $\mu$P (Yang & Hu, 2021; Yang et al., 2022), provides principled rules for zero-shot learning rate transfer across model widths. Thus, large terminal model hyperparameters can be determined by sweeping a small proxy model. $\mu$P has been used to train models up to at least 13B parameters with zero-shot transfer (Blake et al., 2023; Dey et al., 2023; Narayan et al., 2025). Its applicability, however, has been largely limited to transferring learning rates across model widths. To broaden this scope, Dey et al. (2025) introduced Complete-P, extending the original prescriptions to weight decay and model depth. However, many common architectures that are widely deployed in production still lack established $\mu$P scalings.

This paper seeks to close this gap by extending the spectral $\mu$P framework of Yang et al. (2023a) to be more practically useful in deriving $\mu$P prescriptions for novel architectures. As an example of the utility of our framework, we derive (to our knowledge, the first) $\mu$P scaling for grouped-query attention (GQA) (Ainslie et al., 2023). Our analysis reveals that GQA surfaces several difficulties that prior work has left unaddressed. First, when using GQA the original $\mu$P implementation passes coordinate checks, i.e., the customary correctness tests for the implementation. However, empirical analysis shows that the original $\mu$P implementation fails to transfer learning rates, seemingly contradicting established theory (see Figures 2 and 4). We resolve this by extending the spectral-norm version of $\mu$P introduced in Yang et al. (2023a), and showing that the original $\mu$P implementation does not pass a more rigorous spectral-norm coordinate check. Second, the intrinsic low rank of GQA weight matrices skews the expected size of layer outputs. To address this issue, we introduce a new norm, namely the expected operator norm, to replace the spectral norm in spectral $\mu$P theory and restore the desired scaling behavior.

Our primary contributions are threefold:

1. We extend the spectral $\mu$P theory of Yang et al. (2023a), which allows derivations of $\mu$P for more advanced architectures like weight decay, recursion blocks, and GQA. Our work provides, to our knowledge, the first derivation of $\mu$P scaling for GQA.

2. We perform empirical analysis to validate the theory and offer practical guidance for learning rate transfer across GQA settings. In particular, we suggest that transferring across different numbers of GQA repetitions leads to noisy transfer dynamics, suggesting caution when attempting to transfer learning rate.

3. We show with experiments that, with the correct scalings, both weight-decay strength and the training-time constant $\tau_{\text{epoch}}$ introduced in Wang & Aitchison (2024) appear to be transferable.

## 2 RELATED WORK

**Foundations of $\mu$P:** $\mu$P builds on a series of works by Yang, developing the Tensor Programs framework (Yang, 2019; 2020a;b; Yang & Hu, 2021; Yang et al., 2022; 2023b). This line of work uses random matrix theory to carefully analyze the mathematical properties of neural networks during training, while also demonstrating empirically that these theoretical mathematical approaches remain valuable for real-world deep learning. Within the framework of Tensor Programs, Yang et al. (2022) derives the well-known $\mu$P scaling laws for width under SGD and Adam training. The final paper in the series Yang et al. (2023b) attempts to extend $\mu$P to depth scalings. However, they were unable to extend their results to the case of residual blocks with standard configurations for the hidden layers. Finally, the foundation of the mathematical framework presented in this work builds on Yang et al. (2023a), who show an alternative derivation of the results in Yang et al. (2022) based on spectral norms.

**Models using GQA:** Many modern LLM models use grouped-query attention, including LLaMA 2 (Touvron et al., 2023), IBM Granite Mishra et al. (2024), and Mistral 7B Jiang et al. (2023).

**Extensions of $\mu$P:** The original $\mu$P formulation presented in Yang et al. (2022) applies only to scaling the width of a fixed depth, fixed batch size neural network. While already a powerful tool, later authors have sought to extend the principles of $\mu$P to cover cases not covered by the original formulation. Dey et al. (2023) do large-scale validation experiments using $\mu$P and find empirical evidence that learning rate can transfer across batch and dataset size. Dey et al. (2023) suggests $\mu$P-type scalings for weight decay, the Adam $\varepsilon$, and depth. Their contributions to depth scaling are most notable, as their empirical findings contradict the scaling presented in Yang et al. (2023b). However, their extensive empirical analysis suggests that the scaling they derive is correct. We arrive at the same scaling in Section 3.2 using the framework we outline in this paper.

Blake et al. (2023) apply $\mu$P in the context of large-scale, low-precision LLM training. They use ABC parameterizations to apply the $\mu$P scaling rules while maintaining unit variance for all layers in the network, which they refer to as unit scaling-$\mu$P. Additionally, they empirically validate that learning rate transfer persists across datasets, batch sizes, depths, and training iterations under controlled conditions. Narayan et al. (2025) suggest a different, more simplified version of the unit scaling-$\mu$P which they also show works for training low-precision networks with $\mu$P.

## 3 DERIVING NOVEL MAXIMAL UPDATE PARAMETERIZATIONS

Consider a collection of weight matrices $\boldsymbol{W}^\ell \in \mathbb{R}^{n_\ell \times m_\ell}$ in a neural network, indexed by layer $\ell$. Yang et al. (2023a) proves that conditions imposed upon the weight matrices of a network imply feature learning (and thus learning rate transfer) as defined in Yang et al. (2022) (see Equation 3). For initial weights $\boldsymbol{W}_0^\ell$ and iterates $\boldsymbol{W}_t^\ell = \boldsymbol{W}_0^\ell + \sum_{k=1}^t \Delta \boldsymbol{W}_t^\ell$, where $\Delta \boldsymbol{W}_t^\ell = \boldsymbol{W}_t^\ell - \boldsymbol{W}_{t-1}^\ell$, Yang et al. (2023a) suggests that both the initialization and the updates must satisfy:

$$\left\| \boldsymbol{W}_0^\ell \right\| = \Theta(\sqrt{n_\ell}/\sqrt{m_\ell}), \qquad \left\| \Delta \boldsymbol{W}_t^\ell \right\| = \Theta(\sqrt{n_\ell}/\sqrt{m_\ell}), \tag{1}$$

where $\left\| \boldsymbol{W} \right\| := \sup_{\left\| x \right\|_2 = 1} \left\| \boldsymbol{W} x \right\|_2$ is the usual spectral (or induced) norm. This spectral perspective on feature learning is powerful, and we introduce three minor but important modifications that enable us to extend the method of Yang et al. (2023a) to cover novel architectures like GQA.

**Analysis Under a New Norm:** The spectral norm can be interpreted as the maximal deformation of an input vector induced by an operator $\varphi : \mathbb{R}^m \to \mathbb{R}^n$. For full-rank operators, such as dense feed-forward layers, random matrix theory shows that the quantitative value of the spectral norm

is attained asymptotically. In the classical case of an $n \times n$ random matrix $A$, we have the sharp asymptotic relation $\|A\| = 2\sqrt{n}$ as $n \to \infty$.

However, for rank-degenerate matrices like those used in GQA, the spectral norm is not attained asymptotically in practice. The reason is that, as shown by Tensor Programs Yang & Hu (2021), the inputs to a GQA layer during training are i.i.d., and therefore, for rank-degenerate matrices, the vectors that cause this "maximal deformation" occur with probability zero! A visualization of this discrepancy can be seen in Figure 1. Instead, we should use a notion of size that reflects the actual deformation encountered during training.

To this end, let $\Omega$ be the probability distribution of the input vectors. We define the **expectation operator norm** as[1]

$$\|\boldsymbol{A}\|_{\mathbb{E},\Omega,p} := \mathbb{E}_{x \sim \Omega} \left[ \frac{\|\boldsymbol{A}x\|_p}{\|x\|_p} \right]. \tag{2}$$

Throughout this paper, we adopt the convention $\|\boldsymbol{A}\|_E = \|\boldsymbol{A}\|_{\mathbb{E},\mathcal{N}(0,1),2}$, where $x \sim \mathcal{N}(0,1)$ has i.i.d. entries. Crucially, when $A$ is square with i.i.d. entries, it has full rank with probability one, and we obtain the asymptotic relationship $\|\boldsymbol{A}\|_{\mathbb{E}} = \Theta(\|\boldsymbol{A}\|)$. A proof is provided in Lemma 2.

**Operator-Norm Focused Feature Learning:** Yang et al. (2023a) shows that constraining the spectral norm of the weight matrices implies feature learning in the sense of Yang & Hu (2021), where feature learning is defined to occur when

$$\|h_0^\ell\|_2 = \Theta(\sqrt{n}), \qquad \|\Delta h_t^\ell\|_2 = \Theta(\sqrt{n}), \tag{3}$$

for all pre-activations $h^\ell$. In particular, Yang et al. (2023a) prove that enforcing condition equation 1 on spectral norms implies equation 3. However, the converse does not hold: feature learning in the sense of equation 3 may still occur even if the weight matrices do not scale according to equation 1.

Consider a hidden layer $h(x) = \boldsymbol{W}x$ with trainable weights $\boldsymbol{W} \in \mathbb{R}^{n \times n}$ and an additional scaling parameter, the number of layers $L > 1$ independent of $n$, for which we want to ensure feature learning as $L \to \infty$. Under proper initialization, i.e. $\|\boldsymbol{W}_0\| = \Theta(1)$, we have $\|h(x)\|_2 = \Theta(\sqrt{n})$ for $x \in \mathbb{R}^n$ with $\|x\|_2 = \Theta(\sqrt{n})$, as required by feature learning. Now suppose the learning rate is set incorrectly, and $\|\Delta \boldsymbol{W}_t\| = \Theta(L^{-\alpha})$ for some $0 < \alpha$. Then the weight update takes the form

$$\Delta h_t = \boldsymbol{W}_t x_t - \boldsymbol{W}_{t-1} x_{t-1} = \Delta \boldsymbol{W}_t x_t + \boldsymbol{W}_{t-1} \Delta x_t.$$

Assuming that these terms do not exactly cancel, and noting that $\|\Delta x_t\|_2 = \Theta(\sqrt{n})$, we have that

$$\|\Delta h_t\| = \Theta(\sqrt{n}(1 + L^{-\alpha})) = \Theta(\sqrt{n}),$$

and thus this layer satisfies feature learning in the sense of equation 3. For GQA, this precise situation arises, and the subtle failure of the terms $\Delta h_t$ to properly scale leads to a failure of learning rate transfer (see Figures 2 and 4 below).

This analysis shows that the spectral condition of equation 1 is a stronger notion of feature learning than equation 3 and we propose using it as the **definition** of feature learning. This perspective has beneficial practical consequences. When doing coordinate checking to validate a $\mu$P implementation (see (Yang et al., 2022)), we found that directly analyzing the weight matrices proves more effective than analyzing only the activations (see Figure 7). This point is discussed further below.

**A Functional Analytic View of Layer-Wise Computation:** Modern machine learning architectures consist of more than dense feed-forward units. Thus, we propose focusing on the computational units of the network rather than specifically focusing on matrices. In the case of dense feed-forward layers, these notions coincide. But for residual layers our perspective offers a more unifying approach. Concretely, we regard a neural network not only as a compositional sequence of matrix multiplications, but as a compositional sequence of abstract, generally non-linear mappings $\varphi^\ell : \mathbb{R}^m \to \mathbb{R}^n$.

---

[1]Technically, the object we define as $\|A\|_{\mathbb{E},\Omega,p}$ is only a seminorm without further constraints on $\Omega$. In particular, if supp $\Omega \neq \mathbb{R}^n$ then it is possible for all random vectors $x \sim \Omega$ to lie in the nullspace of $A$. This edge case does not occur in neural network training.

We suggest that the first part of the spectral condition in equation 1 should be applied to each compositional unit, rather than the matrices themselves. Starting with the end-to-end computation of the network, we recursively apply this condition to all mappings $\varphi^\ell$. In conjunction with requiring that all trainable parameters satisfy both parts of equation 1, this leads to a unified treatment of residual layers which we discuss below.

### 3.1 WEIGHT DECAY

Weight decay is commonly applied in deep learning to stabilize model training dynamics (Loshchilov & Hutter, 2017; Andriushchenko et al., 2023). For concreteness, we focus on AdamW (Loshchilov & Hutter, 2017) in this section, although our framework extends well to other optimizers with weight decay, including MuON with weight decay (Jordan et al., 2024). AdamW modifies the Adam weight update equation 8 by including a weight decay term with the associated weight decay hyperparameter[2] $\lambda > 0$. With the Adam update step defined as $\hat{r}_t$ we have

$$\Delta \boldsymbol{W}_t = -\lambda \eta \, \boldsymbol{W}_t - \eta \hat{\boldsymbol{r}}_t. \tag{4}$$

To ensure that this term scales correctly in the spectral norm, the spectral norm of each of the individual terms must match. Thus

$$|| \, \Delta \boldsymbol{W}_t \, || = \Theta(\lambda \eta \, || \, \boldsymbol{W}_t \, ||) = \Theta(\eta \, || \, \boldsymbol{r}_t \, ||) = \Theta(1).$$

Because $|| \, \Delta \boldsymbol{W}_t \, || = || \, \boldsymbol{W}_t \, ||$, this means that $\lambda \eta = \Theta(\eta \, || \, \hat{\boldsymbol{r}}_t \, ||)$. Recall that $\mu$P tells us that for input layers $\eta = \Theta(1)$, while for hidden and output layers $\eta = \Theta(n)$. Thus, with $\lambda_0$ being our base weight-decay, we have $\lambda^0 = \Theta(1)$ for the input layers, $\lambda^\ell = \Theta(n)$ for the hidden layers, and $\lambda^{L+1} = \Theta(n)$ for output layers.

We can further characterize the dynamics of weight decay under this scaling and explain why any alternative scaling for the weight decay parameter $\lambda$ either fails to yield transferable learning dynamics or collapses to a standard Adam in the limit that $n \to \infty$. For concreteness, consider a hidden layer.

Suppose we set $\lambda^\ell = \lambda_0 n^{1+\delta}$ for some $\delta > 0$. To preserve transferable dynamics, this would require $\eta = \Theta(n^{-1-\delta})$. In this case, for sufficiently large $n$, we have $\Delta \boldsymbol{W}_t \approx -\eta_0 \lambda_0 \, \boldsymbol{W}_t$, since the first term in equation 4 is much bigger than the Adam update term 9. Consequently, the model ceases to receive gradients from the data at large widths and the weights converge to $\boldsymbol{0}$! Conversely, if we choose $\lambda = \lambda_0 n^{1-\delta}$, $0 < \delta < 1$, then $\eta = \Theta(1/n)$ and we have the opposite problem: for sufficiently large $n$, $\Delta \boldsymbol{W}_t \approx -\eta \hat{\boldsymbol{r}}_t$. So the weight decay is effectively ignored! In this case, training reduces to Adam, and AdamW provides no additional benefit.

Recent works by Wang & Aitchison (2024) and Dey et al. (2025) independently derived a similar relationship between learning rate and weight decay using different approaches. In particular, they interpret weight decay as an exponential moving average and argue that, for a fixed number of iterations, the product $\eta \lambda \times \text{iters} = \text{const}$ should remain constant across model sizes. In our experimental setup, we confirm that this relationship holds (see Figure 8).

### 3.2 COMPLETE-P DEPTH SCALING

We now discuss the depth scaling proposed by Dey et al. (2025), which yields a specific scaling for the residual branches of neural networks. Their derivation is motivated by avoiding the "lazy-learning" regime. Using our spectral framework, we arrive at the same scaling, thus uniting the Complete-P depth scaling with the broader principles of the $\mu$P literature.

Consider the stacked hidden layers $G^\ell(x) = x + \beta g^\ell$, where $1 \le \ell \le L$, and $\beta$ a constant independent of $\ell$. In general, we take $x \in \mathbb{R}^n$ so that the constituent function satisfies $g^\ell : \mathbb{R}^n \to \mathbb{R}^n$.

---

[2]We focus on **coupled** weight decay, which is the type of weight decay included in PyTorch (Schaipp, 2024). However, the weight decay introduced in Loshchilov & Hutter (2017) is **decoupled** and given by $\Delta \boldsymbol{W}_t = -\lambda \, \boldsymbol{W}_t - \eta \hat{\boldsymbol{r}}_t$. Our results still apply in this case and prescribe the scaling $\lambda = \lambda_0$, where $\lambda_0$ is the base model weight decay, to ensure the terms all have the same size in norm. In other words, when using decoupled weight decay, the base weight decay term should not scale with model size. See also Dey et al. (2025).

|  | Embed. | Unemb. | Attn. (Q, O) | Attn. (K, V) | Feed-forward |
|---|---|---|---|---|---|
| Init. Var. | 1 | 1 | $1/\sqrt{n}$ | $1/\sqrt{n}$ | $1/\sqrt{n}$ |
| Multiplier | 1 | $1/n$ | 1 | 1 | 1 |
| LR | 1 | 1 | $1/n$ | $(1+\sqrt{r})/(2n)$ | $1/n$ |
| Weight Decay | 1 | 1 | $n$ | $2n/(1+\sqrt{r})$ | $n$ |

Table 1: The table summarizes the parameterization of Transformers with Grouped-Query Attention (GQA), where $n$ denotes the input dimension and $r$ is the number of key-value head repetitions. Modifications specific to GQA are highlighted in blue. The derivations of learning rate and weight decay follow the AdamW implementation in PyTorch.

Let $\boldsymbol{G} := \bigcirc_{\ell=1}^{L} G^\ell$ denote the $L$-fold composition of these layers. Since we require each individual compositional unit of the neural network to satisfy equation 1, we may restrict attention to the residual block $\boldsymbol{G}$ without loss of generality. Importantly, assuming that the inputs to $\boldsymbol{G}$ satisfy $\|x\|_2 = \Theta(\sqrt{n})$, it follows that for every $\ell$, $G^\ell$ and $g^\ell$ both satisfy equation 1.

We further assume that $\|g^\ell\| = \Theta(1) = \|\Delta g^\ell\|$ with respect to all parameters subject to scaling (width $n$ and depth $L$). Finally, without loss of generality, we restrict to $L \geq 2$, as the single-layer case yields only trivial bounds.

Define $\overline{G}_t^\ell = \bigcirc_{k=1}^\ell G_t^\ell$. In this notation, we have the identity $\overline{G}_t^\ell = \overline{G}_t^{\ell-1} + \beta g_t^\ell \circ \overline{G}_t^{\ell-1}$. These two terms are correlated but only entry-wise, with entries in each matrix i.i.d. according to Tensor Programs (Yang & Hu, 2021). Assuming $g_t^\ell$ is full-rank and $\beta\|g_t^\ell\| < 1$, it follows that $G_t^{\ell-1}$ is also full-rank. Further assuming no exact cancellation, we obtain

$$\left\|\overline{G}_t^\ell\right\| = \Theta\left(\left\|\overline{G}_t^{\ell-1}\right\| + \beta\|g_t^\ell\|\left\|\overline{G}_t^{\ell-1}\right\|\right).$$

With $\beta < 1$, we can use a simple recursive argument to obtain the depth-dependent bound $\left\|\overline{G}_t^\ell\right\| = \Theta(1 + \ell\beta)$. This bound is consistent with the recursion under the assumptions $\left\|\overline{G}_t^0\right\| = \|I\| = 1$ and $\|g_t^\ell\| = 1$. Thus, choosing $\beta = \Theta(L^{-1})$ ensures that $\left\|\overline{G}_t^\ell\right\| = \Theta(1)$ for all layers $1 \leq \ell \leq L$, satisfying the proposed framework. Moreover, Yang et al. (2023b) proves that any exponent $\alpha > 1$ (i.e. $\beta = \Theta(L^{-\alpha})$) leads to trivial dynamics in the limit, so we have arrived at the same bound as Dey et al. (2025) using alternate methods.

### 3.3 GROUPED QUERY ATTENTION

Grouped query attention (GQA) reduces computational cost by repeating the key and value heads in the Transformer (Ainslie et al., 2023). In a standard multi-headed attention layer, the key and value projections are given by weights $\boldsymbol{W}_K \in \mathbb{R}^{n \times n}$ and $\boldsymbol{W}_V \in \boldsymbol{R}^{n \times n}$, where $n$ is the embedding dimension. These matrices are partitioned into $H$ heads of size $n/H$ each (note that $n/H$ must be an integer) and the $i$-th head is computed as $k_i = (\boldsymbol{W}_K x)_i$, $v_i = (\boldsymbol{W}_V x)_i$. In GQA, the number of parameters is reduced by using only $p$ distinct key and value heads, where $H/p = r$ is an integer representing the number of repetitions of each of the $p$ key and value heads. We then define matrices $\boldsymbol{W}_{p,K}, \boldsymbol{W}_{p,V} \in \mathbb{R}^{\frac{n}{r} \times n}$, and construct the full key and value weights by concatenating along the output dimension

$$\boldsymbol{W}_K^\oplus = \bigoplus_{m=1}^r \boldsymbol{W}_{p,K}, \qquad \boldsymbol{W}_V^\oplus = \bigoplus_{m=1}^r \boldsymbol{W}_{p,V}, \tag{5}$$

where $\oplus$ denotes concatenation along the first dimension[3].

Consider the initial weight matrix $\boldsymbol{W}_0$ for either the key or value projections, and its concatenation version $\boldsymbol{W}_0^\oplus$, and let $\boldsymbol{W}_t$ and $\boldsymbol{W}_t^\oplus$ denote their corresponding weight updates. To begin, applying

---

[3]Note that concatenation and matrix multiplication commute: if $\boldsymbol{A} \in \mathbb{R}^{m \times n}$ and $x \in \mathbb{R}^n$, we have $\boldsymbol{A}^\oplus x = (\boldsymbol{A}x)^\oplus$, which follows directly by writing the product in it's index form.

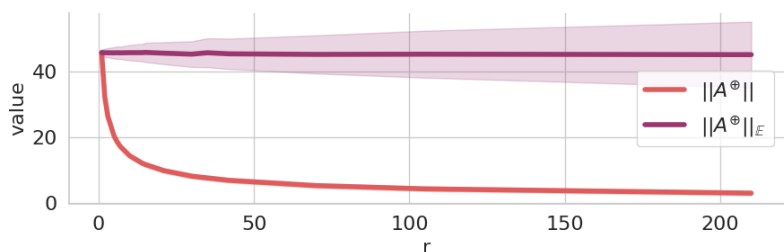

Figure 1: Demonstration of the failure of the spectral norm to accurately capture the behavior for low-rank matrices when the inputs are randomly sampled i.i.d. from $\mathcal{N}(0,1)$. We consider matrices $\boldsymbol{A}$ with entries sampled from $\mathcal{N}(0,1)$ and plot the result of computing $\boldsymbol{A}x$ where the input is a random vector. The top line is $\|\boldsymbol{A}^{\oplus}x\|_2 / \|\boldsymbol{A}^{\oplus}\|_{\mathbb{E}}$ and the bottom line is $\|\boldsymbol{A}^{\oplus}x\|_2 / \|\boldsymbol{A}^{\oplus}\|$. The $x$-axis is the number of repetitions (concatenations) of the matrix $\boldsymbol{A}^{\oplus}$, ranging from 1 to 210 and consisting of all unique factors of 210, the $y$-axis is the measured value of the respective function. Each data point is averaged over 30 such trials to approximate the actual behavior as seen during training. We observe that the expected operator norm is the "correct" scaling for this situation.

the law of large numbers and the central limit theorem to equation 2, we obtain

$$\|\boldsymbol{W}_0^{\oplus}\|_{\mathbb{E}} = \mathbb{E}_x \left[ \Theta \left( \frac{\left( \sum_{k=1}^n \left( \sum_{j=1}^n (W_0^{\oplus})_{kj} x_j \right)^2 \right)^{1/2}}{\left( \sum_{k=1}^n x_j^2 \right)^{1/2}} \right) \right]$$

$$= \mathbb{E}_x \left[ \Theta \left( \frac{\left( r \sum_{k=1}^{n/r} \left( \sum_{j=1}^n (W_0)_{kj} x_j \right)^2 \right)^{1/2}}{\left( \sum_{k=1}^n x_j^2 \right)^{1/2}} \right) \right]$$

$$= \Theta \left( \frac{(r \times \frac{n}{r} \times n \times \sigma^2)^{1/2}}{n^{1/2}} \right) = \Theta \left( \sigma n^{1/2} \right),$$

Thus, to satisfy the spectral condition in equation 1, we require $\sigma = \Theta(n^{-1/2})$. Importantly, this corresponds to the expected operator norm for $\boldsymbol{W}^{\oplus}$, not the spectral norm of the constituent matrix $\boldsymbol{W}$. Because $\boldsymbol{W}_0$ has full rank with probability 1, its spectral norm can be computed directly using Bai-Yin (Bai & Yin, 1993; Yin et al., 1988)

$$\|\boldsymbol{W}_0\| = \Theta \left( \sigma \left( \sqrt{n} + \frac{\sqrt{n}}{\sqrt{r}} \right) \right) = \Theta \left( \frac{1 + \sqrt{r}}{\sqrt{r}} \right). \tag{6}$$

Moreover, in terms of spectral norms, $\|\boldsymbol{W}_0^{\oplus}\| = \sqrt{r} \|\boldsymbol{W}_0\|$ (Lemma 1), so that the spectral norm and the expected operator norm do not agree in this setting (see Figure 1).

The computation in equation 6 is critical for determining the required learning rate, since we require $\|\Delta\boldsymbol{W}_t\| = \Theta(\|\boldsymbol{W}_0\|)$. To this end, we compute $\eta$ in the usual manner. Assuming the use of the Adam optimizer with update step $\hat{\boldsymbol{r}}_t$, we have

$$\|\Delta\boldsymbol{W}_t\| = \eta \|\hat{\boldsymbol{r}}_t\| = \Theta \left( \frac{\eta n}{\sqrt{r}} \right) = \Theta \left( \frac{1 + \sqrt{r}}{\sqrt{r}} \right).$$

From this we easily deduce that $\eta = \Theta \left( \frac{1 + \sqrt{r}}{n} \right)$. We normalize by a factor of two to ensure that when $r = 1$ our scalings agree with the usual full-rank hidden layer scalings:

$$\sigma = \frac{1}{\sqrt{m}} \sigma_0, \qquad \eta = \frac{1 + \sqrt{r}}{2m} \eta_0. \tag{7}$$

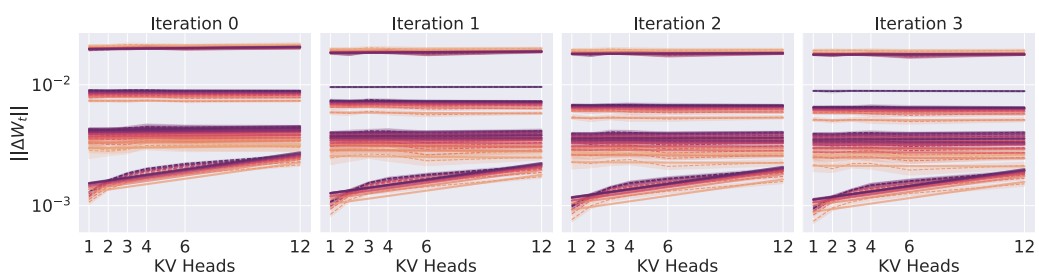

Figure 2: Coordinate checks for $||\Delta \boldsymbol{W}||$ under the vanilla Adam-$\mu$P scalings. The model fails the coordinate checks when evaluated using the spectral feature learning condition equation 1. However, as shown in Figure 7, it does pass when evaluated under Yang's definition of feature learning 3. Further experimental details can be found in Appendix B.1.1.

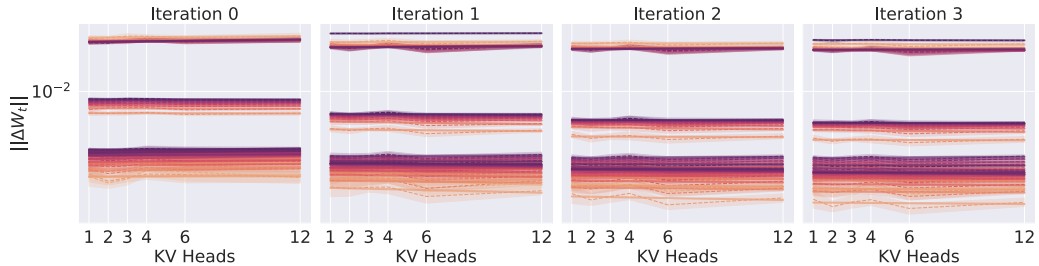

Figure 3: Coordinate checks for $||\Delta \boldsymbol{W}||$ under our proposed GQA scalings. The model has eight hidden layers. Additional experimental details are provided in Appendix B.1.1.

## 4 EMPIRICAL RESULTS

In this section, we present our empirical results. Details of model configurations and experimental setups are provided in Appendix B.1.

**Coordinate-Checks Demonstrate the Necessity for Spectral Feature Learning:** As discussed in Section 3, validating feature learning by measuring the norms of $h^\ell$ and $\Delta h_t^\ell$ can be misleading. Figures 6 and 7 plot $||h_t^\ell||$ and $||\Delta h_t^\ell||$, respectively, for the vanilla Adam-$\mu$P implementation. These coordinate checks would suggest transferable learning rates, yet empirical results show otherwise (see Figure 4, middle). By contrast, when we instead examine the spectral norm conditions in equation 1 (Figure 3), the model fails the coordinate check: a clear non-linear dependence on the number of kv heads of the model, which explains the lack of transferrable dynamics.

**Coordinate-Checks Demonstrate a Qualitative Dependency on $r$:** Because the vanilla Adam-$\mu$P implementation and our implementation share the same initialization scaling, we do not compare $||\boldsymbol{W}||$ directly. Instead, Figure 2 presents the coordinate checks for the vanilla Adam-$\mu$P implementation, while Figure 3 shows the corresponding coordinate checks for our proposed GQA scaling. Our method passes the coordinate check, thereby enabling $\mu$-transfer of learning rate. By contrast, the vanilla Adam-$\mu$P implementation shows a persistent dependency on the number of KV heads, explaining why the learning rate does not transfer in this case.

**Learning Rate Transfer for Grouped Query Attention:** We perform an ablation study comparing the standard parameterization, the vanilla Adam-$\mu$P implementation (where the KV heads are initialized as hidden layers), and our proposed GQA-$\mu$P. The results of this ablation study are summarized in Figure 4. We observe that the vanilla Adam-$\mu$P scaling does not account for the shift induced by using GQA, whereas our proposed scaling brings the optimal learning rates into a much narrower region. Noise inherent to GQA training is already evident in these plots and becomes more pronounced as the number of KV heads decreases. This noise is apparent in both the coordinate checks from Figure 3 as well as in Figure 1. We provide an explanation for this phenomenon in the following paragraph.

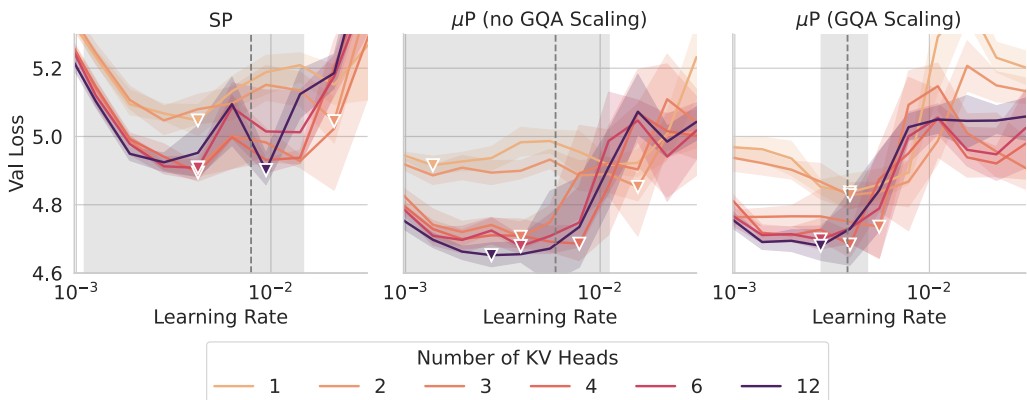

Figure 4: Comparison of the standard parameterization (left), the vanilla Adam-$\mu$P parameterization (middle), and our GQA-$\mu$P scaling (right). For a fixed model size, we vary the number of KV heads. The dashed lines indicate the mean optimal learning rates for each parameterization, and the shaded grey region denotes the standard deviation of the optimal learning rates. All models are trained to 10 tokens per parameter (TPP). Additional details can be found in Appendix B.1.2.

**Expected Variance in GQA Transfer:** From the perspective of $\mu$P, the nature of GQA introduces a dichotomy: one may achieve feature learning in the sense of equation 1 and thereby obtain learning rate transfer, but at the cost of increasingly noisy dynamics as the number of KV heads decreases; alternatively, one may constrain the variance as we decrease the number of kv heads, to stabilize the training, but this leads to a shift in optimal learning rate. Consequently, we suggest that in scenarios where transferable dynamics are critical, it may be preferable to avoid using GQA altogether.

**$\mu$P (Mostly) Decouples Coupled Weight Decay:** To examine the transferability of optimal learning rate and optimal weight decay across model scales, we do a random grid search over (learning rate, weight decay) pairs at constant initial standard deviation. We plot our results in Figure 5. First, we note that under the standard parameterization, neither the learning rate nor weight decay transfers, and that the qualitative properties of the Voronoi-interpolated loss landscape change markedly as the model size increases from 26M to 177M non-embedding parameters. By contrast, both the vanilla Adam–$\mu$P implementation and our proposed scaling preserve their qualitative properties across model sizes.

For the experiment in Figure 5, we quantify the degree of transfer in Table 5 below. We find that the variance of both the optimal learning rate and the optimal weight decay across model sizes is lower for our implementation than for the vanilla Adam-$\mu$P baseline. Thus, it suggests that our proposed implementation enables the transfer of both learning rate and weight decay across model scales, both qualitatively and quantitatively.

Previous work have argued that the quantity $\tau_{\text{epoch}} = (\lambda_0 \times \eta_0 \times \text{iters})^{-1}$ should transfer instead of weight decay (Wang & Aitchison, 2024; Bergsma et al., 2025). We found that both weight decay and $\tau_{\text{epoch}}$ transfer in our experimental setting. This is a non-trivial observation since we vary the number of iterations based on the model size. Figure 8 presents the analog of our interpolation diagram Figure 5 and Table 6 reports the quantitative variance results for $\tau_{\text{epoch}}$. We find that $\tau_{\text{epoch}}$ transfers slightly better than weight decay in our setting.

# 5 CONCLUSIONS

In this paper, we introduced a novel extension of the spectral $\mu$P framework originally developed by Yang et al. (2023a). We can apply our framework to rederive the Complete-P weight decay and depth scalings from Dey et al. (2025). Additionally, we use our framework to derive, for the first time, the $\mu$P scalings for grouped query attention (Ainslie et al., 2023). We perform empirical validation in two directions for our work. First, we explore the empirical nature of learning rate transfer for GQA. We find that we can either do noisy learning rate transfer or fail to transfer the learning rate. This dichotomy is a consequence of the competing scalings between the spectral norm and the expected

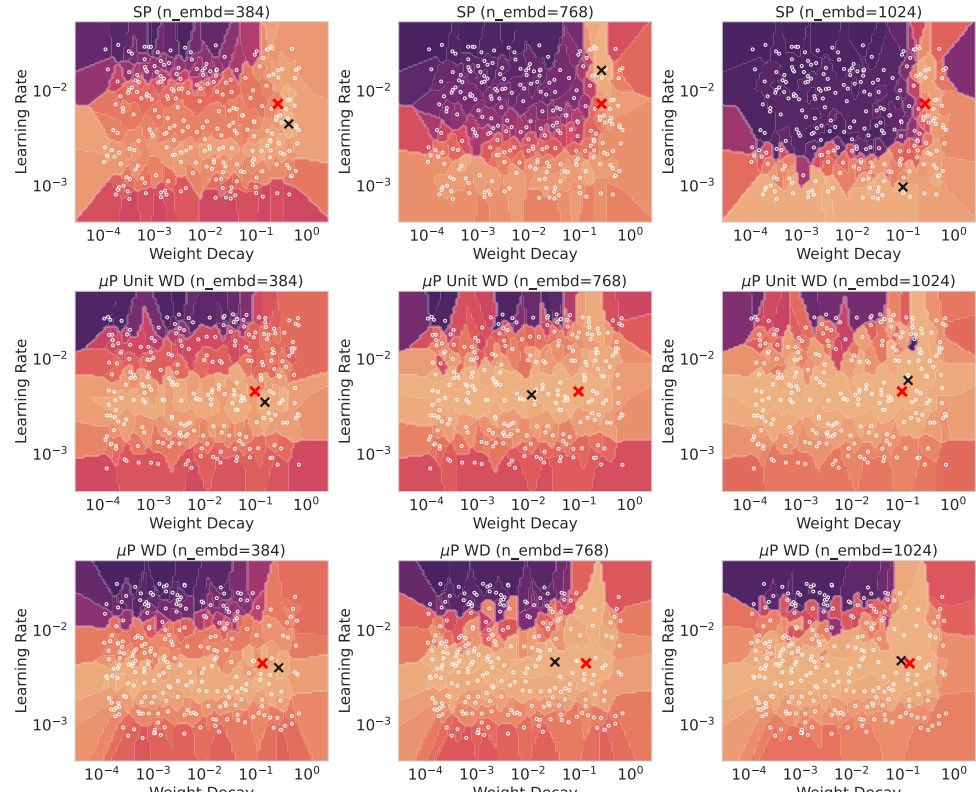

Figure 5: Voronoi interpolation for random sweeps over both learning rate and weight decay. The top row is standard parameterization. The middle row is the vanilla Adam-$\mu$P implementation suggested in Yang et al. (2022). The bottom row is our proposed implementation. Each column corresponds to a different size model, with the number of parameters increasing from left to right. For each model and implementation, we plot the best trial. Hidden dimension, depth, batch size, and training iterations are all scaled. Further details about our training setup can be found in Appendix B.1.3. Lighter colors indicate lower loss, darker colors indicate higher loss. The red crosses mark the average (`learning rate`, `weight decay`) pair, where each coordinate is averaged over the model sizes, while the black crosses are the optimal pair for each experiment.

operator norm. Compared to the standard $\mu$P implementation, our method reduces the variance in optimal learning rate during learning rate transfer. Second, we explore the transferability of weight decay across model sizes. We demonstrate that with the standard $\mu$P implementation, we can nearly achieve transfer of weight decay. With our implementation, we are able to get much closer to true transfer across both learning rate and weight decay.

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
