# A   ADDITIONAL MATHEMATICAL DETAILS

## A.1   DERIVATION FOR ADAM

We demonstrate the applicability of our framework by re-deriving the $\mu$P scalings for Adam. Recall that the Adam optimizer Kingma & Ba (2014) uses hyperparameters $\beta_1$, $\beta_2$, $\varepsilon$, and $\eta$ and has its optimization steps given by the following components:

$$g_t = \nabla_{\boldsymbol{W}} f(\boldsymbol{W}_{t-1}),$$
$$m_t = \beta_1 m_{t-1} + (1 - \beta_1) g_t, \qquad v_t = \beta_2 v_{t-1} + (1 - \beta_2) g_t^2,$$
$$\hat{m}_t = \frac{m_t}{1 - \beta_1^t}, \qquad\qquad \hat{v}_t = \frac{v_t}{1 - \beta_2^t},$$

with the weight update

$$\boldsymbol{W}_t = \boldsymbol{W}_{t-1} - \eta \frac{\hat{m}_t}{\sqrt{\hat{v}_t} + \varepsilon}. \tag{8}$$

The key observation is that the term

$$\hat{\boldsymbol{r}}_t := \frac{\hat{m}_t}{\sqrt{\hat{v}_t} + \varepsilon} \tag{9}$$

will always have typical size 1 (for $\varepsilon$ sufficiently small), and as such the spectral norm can be estimated using the Bai-Yin theorem (Yin et al., 1988; Bai & Yin, 1993), depending on whether the layer is vector-like or matrix-like. Thus, we have the following reasoning. For an input layer, we have

$$\left\| \Delta \boldsymbol{W}_t^0 \right\| = \underbrace{\Theta(\eta^0 \sqrt{n})}_{\text{Bai-Yin}} = \underbrace{\Theta(\sqrt{n})}_{\text{equation 1}},$$

which implies that we must choose $\eta^0 = \Theta(1)$. Next, for the hidden layers, we have

$$\left\| \Delta \boldsymbol{W}_t^\ell \right\| = \underbrace{\Theta(\eta^\ell n)}_{\text{Bai-Yin}} = \underbrace{\Theta(1)}_{\text{equation 1}},$$

which leads us to choose $\eta^\ell = \Theta(n^{-1})$. Finally, for the output layer, we have

$$\left\| \Delta \boldsymbol{W}_t^{L+1} \right\| = \underbrace{\Theta(\eta^{L+1} \sqrt{n})}_{\text{Bai-Yin}} = \underbrace{\Theta(n^{-1/2})}_{\text{equation 1}},$$

which leads us to choose $\eta^{L+1} = \Theta(n^{-1})$. There is a subtle nuance in our derivation that is also often overlooked in the literature. We have assumed that the Adam optimizer step is independent of the network width $n$, but this is not quite true. To see why, consider setting $\beta_1 = \beta_2 = 1$, so that the Adam optimizer step is given simply by

$$\hat{\boldsymbol{r}}_t = \frac{g_t}{|g_t| + \varepsilon},$$

where $g_t$ is the gradient. For concreteness, consider a hidden layer. Yang et al. (2022) show that the gradient will scale like $\Theta(1/n)$. Thus letting $\overline{g}_t = g_t/n$ be the size 1 normalized gradient updates, we have that the

$$\hat{\boldsymbol{r}}_t = \frac{\overline{g}_t}{|\overline{g}_t| + n\varepsilon},$$

which is not actually $\Theta(1)$ in $n$, since for $n = \Omega(\varepsilon^{-1})$, the Adam updates decay like $n^{-1}$. Thus, to be pedantic and ensure actual feature learning, we must scale $\varepsilon = \varepsilon_0/n$ to continue to achieve feature learning. In practice, we find that this subtlety can be avoided by setting $\varepsilon = 10^{-12}$ instead of the usual default of $10^{-8}$; however, for a complete treatment, this scaling must be included. We note that Dey et al. (2023), Everett et al. (2024) make the same conclusion about scaling the Adam $\varepsilon$ parameter and performs an empirical study on its transferability.

## A.2 ADDITIONAL DETAILS FOR GROUPED QUERY ATTENTION

**Lemma 1.** *Scaling Concatenated Spectral Norms. Let $A \in \mathbf{R}^{n \times \frac{n}{r}}$ for some integer $r \geq 1$, where $r$ is the number of repetitions of key and value heads, be a matrix with spectral norm $\|A\| > 0$. Then letting $A^{\oplus} := \bigoplus_{j=1}^{r} A$ we have*

$$\left\| A^{\oplus} \right\| = \sqrt{r} \left\| A \right\|. \tag{10}$$

*Proof of Lemma 1.* We denote the $r$-times concatenation by

$$A^* = [\underbrace{A\,A\,\cdots\,A}_{r \text{ times}}]. \tag{11}$$

Each matrix $A$ has a singular value decomposition $U\Sigma V^T$ for $U, V$ unitary with $U \in \mathbb{R}^{n \times n}$, $V \in \mathbb{R}^{\frac{n}{r} \times \frac{n}{r}}$, and $\Sigma = \begin{bmatrix} \Lambda \\ 0 \end{bmatrix} \in \mathbb{R}^{n \times \frac{n}{r}}$ with $\Lambda$ a diagonal $\mathbb{R}^{\frac{n}{r} \times \frac{n}{r}}$ matrix. Substituting the SVD into equation 11 we can factor out the unitary matrix $U$ and arrive at

$$A = U\,[\,\Sigma V^T\,\Sigma V^T\,\cdots\,\Sigma V^T\,] = U \begin{bmatrix} \Lambda V^T & \Lambda V^T & \cdots & \Lambda V^T \\ 0 & 0 & \cdots & 0 \\ \vdots & \vdots & \ddots & \vdots \\ 0 & 0 & \cdots & 0 \end{bmatrix}$$

It remains to find the singular values of this matrix, which give us the spectral norm scaling. To this end, observe that by the unitary of $V$ we have

$$AA^T = U \begin{bmatrix} \Lambda V^T & \Lambda V^T & \cdots & \Lambda V^T \\ 0 & 0 & \cdots & 0 \\ \vdots & \vdots & \ddots & \vdots \\ 0 & 0 & \cdots & 0 \end{bmatrix} \begin{bmatrix} V\Lambda & 0 & \cdots & 0 \\ V\Lambda & 0 & \cdots & 0 \\ \vdots & \vdots & \ddots & \vdots \\ V\Lambda & 0 & \cdots & 0 \end{bmatrix} U^T$$

$$= U \begin{bmatrix} r\Lambda^2 & 0 & \cdots & 0 \\ 0 & 0 & \cdots & 0 \\ \vdots & \vdots & \ddots & \vdots \\ 0 & 0 & \cdots & 0 \end{bmatrix} U^T.$$

Thus, the largest eigenvalue of $AA^T$ is given by $r\lambda_{\max}^2$, with $\lambda_{\max}$ being the largest eigenvalue of $A$, and the desired spectral norm scaling is immediate. $\square$

**Lemma 2.** *Let $A \in \mathbf{R}^{n \times n}$ have i.i.d. entries. Then the for $x \sim \mathcal{N}(0,1)$ with i.i.d. entries, we have that*

$$\left\| \mathbf{A} \right\|_{\mathbb{E}} = \Theta(\|\mathbf{A}\|).$$

*Proof of Lemma 2.* First, note that $\|\mathbf{A}\| = \Theta(\sigma\sqrt{n})$, where $\sigma$ is the variance of the i.i.d. entries of $\mathbf{A}$. Next, observe that we can use the law of large numbers and the central limit theorem to estimate

$$\mathbb{E}\,\|Ax\|_2^2 = \mathbb{E}\sum_i \left( \sum_j A_{ij}x_j \right)^2 = \Theta(\sigma^2 n^2),$$

and the result follows since $\|x\|_2 = \Theta(\sqrt{n})$. $\square$

Table 2: Model configurations for the coordinate check experiments from Figures 2, 3, 6, 7.

| Width | Depth | Num Heads | Head Size | KV Heads | KV Reps |
|-------|-------|-----------|-----------|----------|---------|
| 576 | 8 | 12 | 64 | 1 | 12 |
| 576 | 8 | 12 | 64 | 2 | 6 |
| 576 | 8 | 12 | 64 | 3 | 4 |
| 576 | 8 | 12 | 64 | 4 | 3 |
| 576 | 8 | 12 | 64 | 6 | 2 |
| 576 | 8 | 12 | 64 | 12 | 1 |

Table 3: Model configurations for the GQA transfer experiments from Figure 4.

| | Params | Non-Embd Params | Width | Depth | Num Heads | Head Size | KV Heads | KV Reps | TPP | Dataset Size (Tokens) | Dataset Size (Sequences) | Batch Size (Tokens) | Batch Size (Sequences) | Iterations |
|---|--------|-----------------|-------|-------|-----------|-----------|----------|---------|-----|-----------------------|--------------------------|---------------------|------------------------|------------|
| kvr_t_1 | 125.55 | 80.62 | 768 | 7 | 12 | 64 | 1 | 12 | 10 | 806200000 | 98413 | 262144 | 32 | 3075 |
| kvr_t_2 | 126.23 | 81.31 | 768 | 7 | 12 | 64 | 2 | 6 | 10 | 813100000 | 99255 | 262144 | 32 | 3102 |
| kvr_t_3 | 126.92 | 82 | 768 | 7 | 12 | 64 | 3 | 4 | 10 | 820000000 | 100098 | 262144 | 32 | 3128 |
| kvr_t_4 | 127.61 | 82.69 | 768 | 7 | 12 | 64 | 4 | 3 | 10 | 826900000 | 100940 | 262144 | 32 | 3154 |
| kvr_t_6 | 128.99 | 84.06 | 768 | 7 | 12 | 64 | 6 | 2 | 10 | 840600000 | 102612 | 262144 | 32 | 3207 |
| kvr_t_12 | 133.12 | 88.19 | 768 | 7 | 12 | 64 | 12 | 1 | 10 | 881900000 | 107654 | 270336 | 33 | 3262 |

# B   EXPERIMENTAL DETAILS

## B.1   MODEL CONFIGURATIONS

In our experiments, we train Transformer language models with untied embeddings and GELU Hendrycks & Gimpel (2023) nonlinearity. The batch size is chosen using a data-driven optimal batch size in equation 12 based on the total number of training tokens $n_{tokens}$, where the corresponding sequence length is 8192.

$$B = 0.000733 \times \sqrt{n_{tokens}}. \tag{12}$$

We use a cosine learning rate schedule with warmup. The number of warmup steps follows equation 13, Dey et al. (2025):

$$n_{\text{warmup}} = \min(\text{int}(0.02 * n_{\text{training}}), \text{int}(375e6/(B \times L))), \tag{13}$$

where B is batch size and L is sequence length.

All of our experiments are conducted using the openwebtext dataset (Gokaslan & Cohen, 2019).

### B.1.1   COORDINATE CHECKS

For the coordinate checking we verify that our norms remain stable as we vary the nubmer of kv heads. The specific configurations which we used during the coordinate checks are contained in Table 2. We use weight decay 0 in our coordinate checking experiments, and do all of the computation in `float32`. We used a fixed Adam $\varepsilon$ of $10^{-12}$ and an initial standard deviation of 0.02. Other optimizer settings are set to the defaults of `PyTorch`'s Adam implementation. We perform our experiments on seeds 1 through 10 and plot the average and confidence interval. We use a batch size of 1 and a sequence length of 1024 to ensure quick computation.

### B.1.2   GQA ABLATION EXPERIMENT

We train our GQA ablation models to 10 TPP. The configurations used for this experiment can be found in Table 3. We set the base weight decay to be $\lambda_0 = 0.1$. We use a base Adam $\varepsilon$ of $10^{-9}/n$, where $n$ is the embedding dimension, to match the predicted Adam $\varepsilon$ scaling of Dey et al. (2025). We take three runs for each data point, using seeds 42, 43, 44 for reproducibility.

Table 4: Model configurations for the Weight decay experiments from Figures 5 and 8.

| | Params | Non-Embd Params | Width | Depth | Num Heads | Head Size | KV Heads | KV Reps | TPP | Dataset Size (Tokens) | Dataset Size (Sequences) | Batch Size (Tokens) | Batch Size (Sequences) | Iters. |
|---|---|---|---|---|---|---|---|---|---|---|---|---|---|---|
| jwd-small | 48.82 | 26.38 | 384 | 4 | 6 | 64 | 6 | 1 | 3 | 79140000 | 9661 | 81920 | 10 | 966 |
| jwd-medium | 125.96 | 81.07 | 768 | 6 | 12 | 64 | 12 | 1 | 3 | 243210000 | 29689 | 147456 | 18 | 1649 |
| jwd-large | 237.17 | 177.31 | 1024 | 10 | 16 | 64 | 16 | 1 | 3 | 531930000 | 64933 | 212992 | 26 | 2497 |

### B.1.3 WEIGHT DECAY TRANSFER EXPERIMENT

Due to the high number of sampling points we only trained our models in the weight decay experiments to 3 TPP, well below the compute optimal horizon (Hoffmann et al., 2022). The configurations used for this experiment can be found in Table 4. We uniformly sample the grid in $\log - \log$ space. We were only able to run one trial per data point, but the high number of trials increase confidence. We sample 250 points on the grid for each implementation and model size.

### B.2 FAILURE OF YANG-TYPE COORDINATE CHECKING

Yang et al. (2022) suggest measuring $\|h_t\|_2$ and $\|\Delta h_t\|_2$ to verify that a $\mu$P implementation is correct by comparing these norms during training to feature learning conditions in equation 3. In Figure 7 we plot $\|\Delta h_t\|_2$ for the vanilla Adam-$\mu$P implementation while varying only the number of kv heads. Note that the the implementation passes a coordinate check, but as discussed theoretically in Section 3, and empirically in Section 4 the learning rate does not transfer for this implementation (see Figure 4).

Coordinate checks on our proposed spectral condition from equation 1, however, capture the failure of feature learning (see Figure 2).

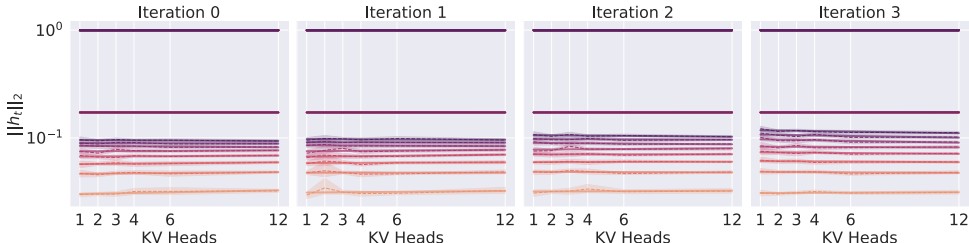

Figure 6: Yang et al. (2022) style coordinate checks on $\|h_t\|_2$. This coordinate check, along with the coordinate check shown in Figure 7, indicate that the implementation is correct, and that feature learning and thus learning rate transfer should occur. However, we show in Figure 4 that learning rate transfer for this implementation does not occur. To understand why, we can look at the failing spectral norm coordinate check in Figure 3.

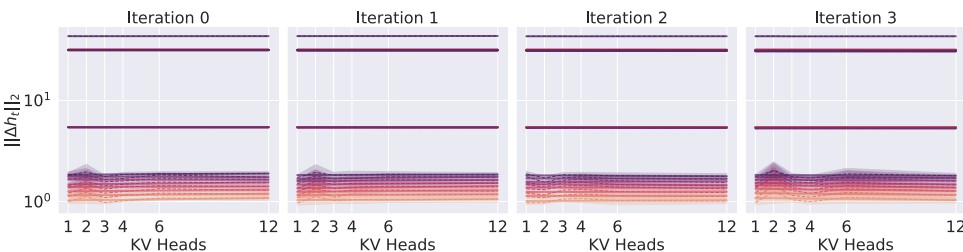

Figure 7: Yang et al. (2022) style coordinate checks on $\|\Delta h_t\|_2$. This coordinate check, along with the coordinate check shown in Figure 6, indicate that the implementation is correct, and that feature learning and thus learning rate transfer should occur. However, we show in Figure 4 that learning rate transfer for this implementation does not occur. To understand why, we can look at the failing spectral norm coordinate check in Figure 3.

### B.3 MORE RESULTS ABOUT WEIGHT DECAY

We used the same data that was collected from Figure 5 to analyze whether or not our experimental testbed demonstrates transfer over $\tau_{\text{epoch}}$, as is suggested by (Wang & Aitchison, 2024; Bergsma et al., 2025; Dey et al., 2025). We find that we get slightly better transfer in $\tau_{\text{epoch}}$ than we do with weight decay, using the same data. We plot the variance in our optimal configurations in Table 6. Like for the case of weight decay transfer (see Figure 5), we find that our suggested implementation outperforms both the standard parameterization and the vanilla Adam-$\mu$P implementation from Yang et al. (2022).

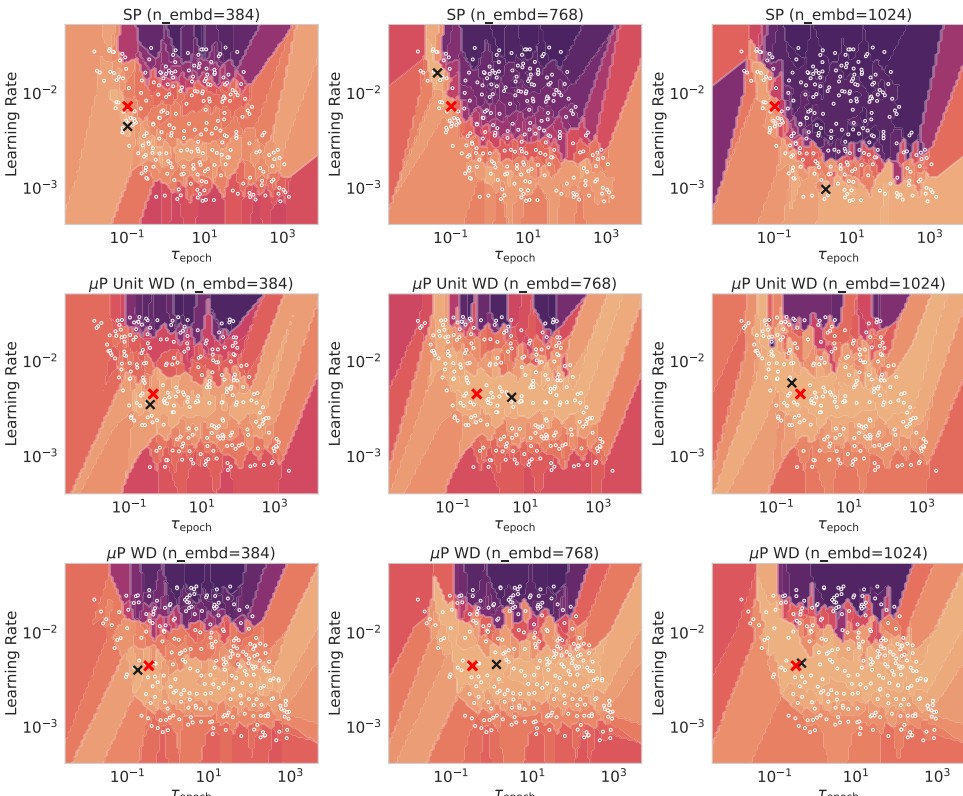

Figure 8: Voronoi interpolation for random sweeps over both learning rate and $\tau_{\text{epoch}}$ Wang & Aitchison (2024). The top row is standard parameterization. The middle row is the vanilla Adam-$\mu$P implementation suggested in Yang et al. (2022). The bottom row is our proposed implementation. Each column is a different size model, increasing in number of parameters from left to right. For each model and implementation we plot the best trial. We scale the hidden dimension, depth, batch size, and training iterations. Further details about our training setup can be found in Appendix B.1.3. Lighter colors are lower loss, darker colors are higher loss. The red x is the average (learning rate, weight decay) pair, where each coordinate is averaged over the model sizes, while the black x is the optimal pair for each experiment.

Table 5: Variance table comparing our implementations across model sizes for the weight decay experiment from Figure 5.

| Implementation | Var. LR | Var. WD | Var. Loss |
|---|---|---|---|
| SP | 1.34 | $3.83 \times 10^{-1}$ | $4.87 \times 10^{-1}$ |
| $\mu$P | $4.75 \times 10^{-2}$ | 1.38 | $4.87 \times 10^{-1}$ |
| $\mu$P + WD | $5.54 \times 10^{-3}$ | $7.51 \times 10^{-1}$ | $4.77 \times 10^{-1}$ |

Table 6: Variance table comparing our implementations across model sizes for the $\tau_{\text{epoch}}$ experiment from Figure 8.

| Implementation | Var. LR | Var. $\tau_{\text{epoch}}$ | Var. Loss |
|---|---|---|---|
| SP | 1.34 | 2.78 | $4.87 \times 10^{-1}$ |
| $\mu$P | $4.75 \times 10^{-2}$ | 1.49 | $4.87 \times 10^{-1}$ |
| $\mu$P + WD | $5.54 \times 10^{-3}$ | $6.56 \times 10^{-1}$ | $4.77 \times 10^{-1}$ |

## C    LLM STATEMENT

We did not use LLMs in a significant way to aid our research during the completion of this work. Our LLM usage did not extend beyond using code assistants like copilot and for polishing the writing in our manuscript.