# OpenReview forum: "GQA-$\mu$P: The Maximal Parameterization Update for Grouped Query Attention and Fully Sharded Data Parallel"
_ICLR.cc/2026/Conference — Submitted to ICLR 2026_

### Official Review · Reviewer_fnei · 2025-10-24

**Soundness:** 4
**Presentation:** 4
**Contribution:** 3
**Rating:** 6
**Confidence:** 2

**Summary:**

The authors consider hyperparameter transfer across model architecture sizes, in particular, how to flexibly adapt the maximal update parameterization for modern architectures which naturally exhibit low rank weight matrices. In order to adapt the framework to such settings, they consider spectral norms conditions as the definition of feature learning, which makes a subtle weakness of the definition via activations apparent. In addition, they define a spectral norm which preserves scaling for sparse matrices by measuring expected transformation rather than maximal transformation as possible with full rank matrices. Together with empirical experiments they show that the hyperparameter transfer works for grouped query attention across the number of repetitions and weight decay.

**Strengths:**

- Interesting contribution to making $\mu$P parameterisation more precise for specific modern architecture types.
- Clear presentation of the intuition and results.
- New derivation of previous work on depth and weight decay scaling using a different method is insightful.
- The idea of having the norm of scaling in expectation seems like it might be useful elsewhere.

**Weaknesses:**

- It is unclear how significant the difference between the vanilla muP scaling and the new method are in practice. The main advantage (as also anticipated from Fig.1) seems to occur for very small $r$. I have difficulty in understanding intuitively whether this is expected or why the problem with the vanilla scaling does not seem to hold for large r anymore. I might miss a point here, so it would be great to clarify this.
- The code is not available.

**Questions:**

Typos:
- L.431 : . .
- Table 1 math mode in some of the lines

---

> ### Author Response · Authors · 2025-12-04
> **Response to Reviewer 4**
>
> Thank you for taking the time to read and review our work.
>
> The reviewer highlighted our clear presentation, novel derivation, and the potential for our techniques to be used in other aspects of deep learning.
>
> With regard to the weaknesses:
> 1. The main advantage will actually be for high $r$, where a single head is copied $r$ times. $r=1$ corresponds to normal attention.
> 1. We are planning to release our code in it's entirety. Our experiments are based on a modified version of NanoGPT.
>
> Thank you again for your review.

---

### Official Review · Reviewer_9sqf · 2025-10-31

**Soundness:** 2
**Presentation:** 2
**Contribution:** 2
**Rating:** 2
**Confidence:** 5

**Summary:**

The paper “GQA-μP: The Maximal Parameterization Update for Grouped Query Attention” extends the theoretical foundation of the maximal update parametrization (μP) by proposing a generalized mathematical framework that introduces the expected operator norm to handle low-rank weight structures, such as those found in Grouped Query Attention (GQA). The authors begin with the standard μP scaling condition on weight matrices $W_\ell \in \mathbb{R}^{n_\ell \times m_\ell}$ given by $||W_\ell^0|| = \Theta(\sqrt{n_\ell/m_\ell})$ and $||\Delta W_\ell^t|| = \Theta(\sqrt{n_\ell/m_\ell})$ They argue that while this condition ensures feature learning for full-rank operators, it fails for rank-deficient matrices like those in GQA, since the spectral norm grossly overestimates the true deformation of random inputs and the event has measure zero, in a simple sense. To resolve this, they define the expected operator norm as $||A||_{E,\Omega,p} := \mathbb{E}{x \sim \Omega} \left[ \frac{||Ax||_p}{||x||_p} \right]$. For full-rank matrices, $||A||_E = \Theta(||A||)$ asymptotically.

Under this proposed norm, with some basic conditions on the input, the paper re-derives feature learning, enforcing the stronger spectral condition directly on weights rather than activations. They extend this framework to include weight decay under AdamW updates $\Delta W_t = -\lambda \eta W_t - \eta \hat{r}_t$, showing that correct scaling requires $\lambda^0 = \Theta(1)$ for input layers and $\lambda^l = \Theta(n)$ for hidden/output layers. The analysis of residual architectures recovers the Complete-P depth scaling, proving that residual strength $\beta = \Theta(L^{-1})$ keeps $||\bar{G}_t^\ell|| = \Theta(1)$ across layers.

As a core contribution, for Grouped Query Attention, where keys/values are repeated $r$ times over $p$ distinct heads ($H/p = r$), the authors derive the corresponding parametrization when the concatenated matrices are used (these concatenated matrices follow the $\sqrt{r}$ scaling for the norm of the corresponding matrix in each group $i, i\in [r]$).

Empirical experiments confirm that this GQA-μP scaling restores learning rate transfer across repetitions and maintains consistent training dynamics, unlike standard μP implementations. The paper thus unifies spectral and expected-norm perspectives, re-derives Complete-P scaling laws, and delivers the first μP-compatible parameterization for GQA, achieving transferable learning rate and weight decay across varying model configurations

**Strengths:**

1. The extension of $\mu$P to shared key value algorithms, such as GQA is certainly a useful addition to the literature, given the exploding size of the models.

2. The Voroni interpolation in Figure 5 clearly demonstrates the strength of this work.

**Weaknesses:**

1. A core reason that I will thoroughly stress – the Related Work section should have been put right after the Intro on the second page. For this work, which heavily compares and references the background work, I believe the current strategy to put the Related Work at the end is perhaps not the best for the reader, irrespective of the fact that the reader is familiar with the literature.

2. The mismatch between the paper's title on Openreview - “GQA-$\mu$ P: The Maximal Parameterization Update for Grouped Query Attention and Fully Sharded Data Parallel” and the one in the actual PDF - “GQA-µP: THE MAXIMAL PARAMETERIZATION UPDATE FOR GROUPED QUERY ATTENTI” is a but troubling. There is no discussion of FSDP across the whole paper.

3. The authors have attached the supplementary section separately, but then also deferred all the experimental details to the supp PDF. What exactly is the reason here?

4. Are the authors sure about Figure 1 on page 6? The denominator seems to be showing norm of the concatenation of A to adjust the rank lower. Should it not have been the norm of the vector “x”? Also, why was each data point averaged, when we are instead looking at max value attained for spectral norm? Additionally, averaging over merely 30 points is too low for such high dimensional cases as considered in the ML setup.

5. I have some questions about the proof of lemma 2 – how is it that the proof simply expands the expectation of the multiplication of the matrix A with the vector x, ie, ||A x|, without considering the numerator of norm of the vector “x” ? Also, the last statement on page 13 about the norm of the vector “x” scaling as $\Theta{\sqrt{n}}$ does not make sense. There should some expectation value in this equation.

6. This statement at the end of page 2 - “For neural network training all distributions have full support.” seems a bit misleading to me. A majority of the data distributions can lie on low-dimensional manifolds, or simply there can be extremely low variance subspaces around the null vector, making it problematic. Further discussions around this point will be helpful.

7. Equation on line 111 is incorrect. Many notational inconsistencies such as equation of $G^l(x)$ on line 181 and equation 5 for the concatenation operation.

**Questions:**

1. What exactly is the definition of “coordinate checks” that the authors mention throughout the paper?

---

> ### Author Response · Authors · 2025-12-04
> **Response to Reviewer 3**
>
> First, thank you to the reviewer for taking the time to read our work and to give us thorough feedback. We appreciate your effort in this process.
>
> ### Weaknesses
> 1. We think the reviewer is correct and have moved the related works sections to the start of the paper.
> 2. We were originally going to cover both topics in a single paper, but decided instead to defer the publishing of our FSDP work to a later date. We forgot to change the title when we uploaded the abstract.
> 3. This is to meet the ICLR page count.
> 4. No, Figure 1 is correct. The spectral norm of the concatenation decays in it's spectral norm like $1/\sqrt{r}$ (see $(10)$ in the Appendix). $\mu$P considers the averaged behavior of a network, not the maximal behavior (an expectation operator is an average). We re-ran with 1000 data points, which took < 45 seconds. We agree with the reviewer that this is a better sample size and have updated the figure.
> 5. $x\sim \mathcal{N}(0, 1)$ so $||x||_2=\Theta(\sqrt{n})$ as stated in the statement of the lemma and the proof.
> 6. Yes, this probably warrants further discussion. The issue is actually much more subtle. In deep learning the distributions behave as if they have full support, even though the may not theoretically have full support. In other words, we do not expect all of the data (or incoming activations) to lie in the nullspace of a weight matrix. We have modified the footnote to reflect this.
> 7. We have fixed the typo on line 111. We are not sure what notational inconsistencies the reviewer is referencing.
>
> ### Questions
> 1. Coordinate checks are the standard method of verifying the implementation of $\mu$P, as discussed in the original $\mu$P paper.

---

### Official Review · Reviewer_U6y7 · 2025-11-01

**Soundness:** 2
**Presentation:** 3
**Contribution:** 2
**Rating:** 4
**Confidence:** 2

**Summary:**

This paper extends the Maximal Update Parameterization (μP) framework to Grouped Query Attention (GQA), enabling zero-shot learning rate transfer across complex Transformer architectures. Building on the spectral feature-learning view of Yang et al. (2023a), the authors redefine feature learning via spectral norms and introduce the expected operator norm to handle low-rank GQA matrices where standard spectral norms fail. This unified spectral μP framework yields the first derivation of μP scalings for GQA, aligning with Complete-P depth and weight-decay principles without relying on lazy-learning theory. Empirical results confirm effective transfer of learning rate and weight decay across GQA repetition factors, though with increased noise and computational cost when the number of key–value head repetitions varies.

**Strengths:**

- The paper provides the first known derivation of μP scalings for Grouped Query Attention (GQA). The derived scalings enable learning rate transfer for GQA models, which the vanilla Adam-μP implementation failed to achieve.
- By relying on the spectral condition on weights as the core definition of feature learning, the authors re-derive the Complete-P depth and weight-decay scalings.  This method unifies the Complete-P extensions with the broader principles of the μP literature.
- The empirical analysis shows that the proposed implementation significantly improves the transferability of both optimal learning rate and optimal weight decay across model scales, qualitatively and quantitatively, compared to the standard μP baseline.

**Weaknesses:**

- Although the paper successfully extends μP, the necessity of introducing a specialized norm—the expected operator norm—highlights the intrinsic difficulty of applying μP to novel architectures. This suggests that deriving μP for future complex architectures may continue to be challenging.
- While the GQA-μP implementation yields better weight decay transfer than vanilla μP, the authors note that the related quantity transfers slightly better. This suggests that the derived weight decay scaling may not be optimally tuned for zero-shot transfer across all training regimes.
- Experiments appear limited to models ≤ 177 M. It is unclear if findings generalize to billions-scale models.

**Questions:**

- Given the observed dichotomy between achieving feature learning transfer and managing increasing noise as the number of KV heads decreases, what is the practical threshold or model size/GQA setting where the accumulated noise negates the benefit of zero-shot learning rate transfer?
- The paper states that the spectral condition is a stronger definition of feature learning than the activation condition. Can the authors provide a more detailed theoretical explanation as to why maintaining the activation norm size is insufficient for guaranteeing transfer stability in this specific rank-degenerate context (GQA)?
- Have you tested the proposed GQA-μP scaling on >1 B-parameter models or on real pre-training datasets to verify zero-shot transfer?

---

> ### Author Response · Authors · 2025-12-04
> **Response to Reviewer 2**
>
> First, we would like to thank the reviewer for taking the time to read and review our work, and to pose some excellent questions to bolster our paper.
>
> ### Weaknesses
> - We point out that Figures 5 and 6 are the same data. So it may be more accurate to say that under our implementation both weight decay and $\tau_{\text{epoch}}$ transfer better than under vanilla-$\mu$P. But we stress that both these quantities transfer.
>
> ### Questions
> 1. This is a fantastic question. We have performed additional experiments since submitting the original paper and we have found that somewhere between 48 and 128 head size (with 12 heads), the shift incurred in the standard parameterization is far more costly than the noise from GQA. As head size is scaled, the noise of GQA is reduced, albeit at a much slower rate than one would like. The experiment we presented in the paper has head size of 48, which contributes to the noise.
> 2. Yes, please see lines 95-118 in the original submission (lines 126-146 in updated submission), and Figure 2. As discussed in the paper, Figures 2, 6, and 7 demonstrate that measuring the layer activations alone is insufficient.
> 3. We have transferred to 750M parameters but have not yet performed an experiment past 1B parameters. We are working on scaling our code to run these experiments and hope to include them in future versions of this work.

---

### Official Review · Reviewer_F9pN · 2025-11-01

**Soundness:** 3
**Presentation:** 3
**Contribution:** 3
**Rating:** 4
**Confidence:** 3

**Summary:**

The paper revisits the spectral norm conditions on the weight evolution in Tensor Programs, and observes conditions where the empirical and analytical checks on the existence of feature learning passes, despite failure to transfer optimal learning rates across width, $n$.
The authors propose an `expectation operator norm` as a more suitable norm for analysing grouped-query attention (GQA) layers that share the KV weights across the Q groups and can thus break the full rank of the GQA weight matrix.
They thus propose a new view for feature learning that encompasses rank degenerate architectural components such as the GQA layer.
The authors also re-establish existing literature findings in the role of the learning rate and weight decay interaction in width scaling being of constant scale, i.e., $\lambda \eta = \Theta(1)$.

**Strengths:**

* Clear motivation and scope.

* Solves a practical gap ($\mu$P for GQA layers) using theory-driven contribution ($||A||_E$).

* Easy to follow notations and math for the most part.

* Reconciliation or rediscovery of previous literature findings (CompleteP depth-scaling; $\tau$ from AdamW's EMA perspective)

**Weaknesses:**

* Despite a practice-driven motivation and problem scope, a lack of empirical evaluations to show the explicit gain in practice or improvement in zero-shot hyperparameter transfer.

* Unclear how the rank degeneracy of certain architectural components (like GQA) affects the learning rate landscape overall, and also convergence rates: L1 checks pass, but LR shifts, but unclear if GQA's low rank enforces a general flatter LR landscape or not.

* Some of the inline math has assumptions and jumps that can do with more referencing or explanations.

**Questions:**

Below is an enumeration of various questions and suggestions.

Please note, the score increase in contingent on the points below, with more weightage on points: 4, 5, 7.

1\. L41-47: Could the authors perhaps refer here to a plot or a table in the paper for the reader?

2\. L111: Shouldn't the second term on the RHS be: $W_{t-1}$ instead of $W$?

3\. L118-134: Hard to follow and the link with the previous and after of this section.

4\. L149-152: Could the authors please break the steps here a bit more?

5\. Is Figure 4 on GQA's only? Given the usual expectation that in SP we typically see a scaling relation with width, which allows $\mu$P to *scale* appropriately with $n$ and align the $LR_0$ at $n$. Could the authors please explain how to interpret this here?

6\. Minor typos:
* L127: *resdiual* $\rightarrow$ residual
* L132: *applying* $\rightarrow$ apply

7\. Figure 5:
* How does one interpret the sizes of the voronoi cells here?
* Do all the plots share the same loss scale and therefore the colors?
* Can we observe the LR landscape wrt width, assuming optimal Weight Decay? (Or how does this play out with a tuned, fixed $\tau$?)
* Does the GQA-scaling appear to have larger regions of loss compared to vanilla-$\mu$P? (top purple region)
* Can there be a learning curve comparison too (at least in the Appendix)?

8\. L643-647: Something that has been explored in [1]?


---

References:

[1] Scaling Exponents Across Parameterizations and Optimizers, Everett et al., 2024, arXiv:2407.05872 [cs.LG].

---

> ### Author Response · Authors · 2025-12-04
> **Response to Reviewer 1**
>
> First, we would like to thank the reviewer for their time, effort, and detailed review. We appreciate the time that you clearly dedicated to our work.
>
> The reviewer highlights the clear motivation of our work and the value of our proposed solution.
>
> ### Answers to Questions
> 1. Yes.
> 2. Yes, thank you.
> 3. Yes, we have re-worked this paragraph.
> 4. To preserve training dynamics, the weight decay term ($\lambda \eta W_t$) must scale proportionally to the weight matrix itself in the spectral norm. That is, we require $\|\lambda \eta W_t\| = \Theta(\|W_t\|)$. Since $\|\lambda \eta W_t\| = \lambda \eta \|W_t\|$, the condition simplifies to $\lambda \eta = \Theta(1)$. For hidden layers, the standard $\mu$P learning rate scales as $\eta = \Theta(1/n)$ (as correctly listed in Table 1).  Substituting $\eta = \Theta(1/n)$ into the condition $\lambda \eta = \Theta(1)$ yields $\lambda (1/n) \approx 1$, which implies $\lambda = \Theta(n)$.
> 5. Thank you for this question. Yes, just GQA. Left (SP): We vary the number of KV heads (repetition $r$) for a fixed model size. In SP, the optimal learning rate shifts significantly as we change the GQA configuration because SP does not account for the change in the operator norm induced by $r$. Middle (Vanilla $\mu$P): Standard $\mu$P also fails to align the curves because it uses the spectral norm, which overestimates the matrix size for low-rank GQA (as shown in Fig 1). Right (GQA-$\mu$P): Our scaling accounts for $r$, bringing the optimal learning rates into a narrow, aligned region.
> 6. Thank you.
> 7. Sure
>     - The Voronoi cells represent the nearest-neighbor interpolation from our random grid search over (LR, WD) pairs. Larger cells simply indicate regions where the random sampling was sparser; the color indicates the loss.
>     - Yes, lighter colors indicate lower loss and darker colors indicate higher loss consistently across plots.
>     - We're not entirely sure what the reviewer means by this. The learning rate landscapes with respect to width are show in the original $\mu$P paper, our implementation can reproduce these results.
>    - The ``purple region" (high loss/divergence) in GQA-$\mu$P (bottom row) is consistent across widths, whereas in SP (top row), the shape of the landscape changes markedly. The consistency of the region is the key indicator of successful transfer.
>     - We will add learning curves to the Appendix in the final version.
> 8. Yes, we were aware of this paper and it should have been included. Thank you.

---

### Meta-Review · Area_Chair_nHsX · 2025-12-16

**Summary:**

This paper extends the theoretical foundation of the maximal update parametrization (μP) by proposing a generalized mathematical framework that introduces the expected operator norm to handle low-rank weight structures, such as those found in Grouped Query Attention (GQA). The derived scalings enable learning rate transfer for GQA models, which the vanilla Adam-μP implementation failed to achieve. However, most of the reviewers complain about a lack of empirical evaluations, validality, significance. The rebuttal didn't answer it very well. Hence, I suggest to reject this paper.

**Reviewer Concerns:**

Reviewer F9pN complains about a lack of empirical evaluations, less explainations. But the weakness were not fully addressed in the rebuttal.

Apart from the limited evaluations, Reviewer U6y7 pointed out that the derived weight decay scaling may not be optimally tuned for zero-shot transfer across all training regimes. The rebuttal didn't address very well.

Reviewer 9sqf mentioned several logisitics issues and the writing style. The rebuttal addressed it well.

Reviewer fnei is unclear about how significant the difference between the vanilla muP scaling and the new method are in practice. The rebuttal is clear.

**Reviewer Scores:**

The rebuttal didn't address the reviewers' concerns very well. Reviewer F9pN, U6y7 would not change their score. Reviewer 9sqf would increase the score. Reviewer fnei has provided positive support.

---

### Decision · Program_Chairs · 2026-01-26

Reject